# Effect of Acoustic Oscillations on Non-Equilibrium State of Magnetic Domain Structure in Cubic Ni_2_MnGa Single Crystal

**DOI:** 10.3390/ma16072547

**Published:** 2023-03-23

**Authors:** Anxo Fernández González, Konstantin Sapozhnikov, Pavel Pal-Val, Sergey Kustov

**Affiliations:** 1Departament de Física, Universitat de les Illes Balears, Cra Valldemossa km 7.5, 07122 Palma de Mallorca, Spain; 2Solid State Physics Division, Ioffe Institute, Politekhnicheskaya 26, 194021 St. Petersburg, Russia; 3B. Verkin Institute for Low Temperature Physics and Engineering of NAS of Ukraine, Nauky Ave. 47, 61101 Kharkiv, Ukraine

**Keywords:** magnetic shape memory alloys, ultrasonics, internal friction, reversible Villari effect, magnetic domain walls

## Abstract

Magnetic hysteresis is a manifestation of non-equilibrium state of magnetic domain walls trapped in local energy minima. Using two types of experiments we show that, after application of a magnetic field to a ferromagnet, acoustic oscillations excited in the latter can “equilibrate” metastable magnetic domain structure by triggering the motion of domain walls into more stable configurations. Single crystals of archetypal Ni_2_MnGa magnetic shape memory alloy in the cubic phase were used in the experiments. The magnetomechanical absorption of ultrasound versus strain amplitude was studied after step-like changes of a polarizing magnetic field. One-time hysteresis was observed in strain amplitude dependences of magnetomechanical internal friction after step-like variations of a polarizing field. We distinguish two ingredients of the strain amplitude hysteresis that are found in the ranges of linear and non-linear internal friction and show qualitatively different behavior for increasing and decreasing applied polarizing fields. The uncovered effect is interpreted in terms of three canonical magnetomechanical internal friction terms (microeddy, macroeddy and hysteretic) and attributed to “triggering” by acoustic oscillations of the irreversible motion of domain walls trapped in the metastable states. To confirm the suggested interpretation we determine the coercive field of magnetization hysteresis through the measurements of the reversible Villari effect. We show that the width of the hysteresis loops decreases when acoustic oscillations in the non-linear range of domain wall motion are excited in the crystal. The observed “equilibration” of the magnetic domain structure by acoustic oscillations is attributed to the periodic stress anisotropy field induced by oscillatory mechanical stress.

## 1. Introduction

Rearrangements of domain structure are behind all useful properties of ferroic materials. Therefore, studying the dynamics of domain walls (DW) is of a prime practical and theoretical importance. Additional interest in DWs is driven by their potentially unusual properties, different from the properties of the bulk. Examples are superconductive twin boundaries in WO_3_ [1], polar twins in non-polar oxide ferroelastics such as CaTiO_3_ [2] and polar tweed in LaAlO_3_ [3]. Designing specific DW properties and configurations for their applications as devices is referred to as “domain wall engineering” [4]. One more fundamental feature of ferroics is the hysteretic response to applied periodic fields. The phenomenon of hysteresis has been the subject of intense theoretical and experimental research [5]. A trajectory along the hysteresis loop represents a sequence of metastable states of the DW system. In ferromagnetics, thermally activated relaxation of DWs trapped in the metastable configurations towards deeper energy minima and closer to equilibrium is known as magnetic viscosity or thermal fluctuation magnetic aftereffect [6,7]. Similar relaxations of domain structures towards equlibrium state are known in ferroelastics, multiferroics, and even antiferromagnets, see e.g., [8,9,10,11]. A flat distribution of activation energies results in the logarithmic kinetics of relaxation [6]. If the activation energies are high, DWs are essentially frozen in the metastable states. Even when thermal activation is inefficient, the transition towards equilibrium under a constant conjugate field can be athermally triggered by an external coupled non-conjugate field. In the case of magnetic DWs, an example of potential perturbation is mechanical stress that affects magnetic domain structure through magnetoelastic coupling [12]. In another scenario, a periodic mechanical stress applied to a conductive ferromagnet in acoustic experiments yields magnetic DW-related softening of elastic constants (modulus defect) and absorption of elastic energy of oscillations (internal friction, IF) [13,14]. Acoustic methods of studying magnetic DW dynamics are highly efficient:while the penetration of a high-frequency magnetic field in the volume of a conductive ferromagnet is essentially limited by the skin depth, mechanical stress can be applied uniformly to the volume of the sample and induce the motion of all non-180° DWs;acoustic methods cover a very wide spatial scale of DW dynamics from much less than magnetic domain width up to the characteristic sample dimension [15,16].

In the present work we combine these two functionalities of the mechanical stress applied to a ferromagnet. We show that acoustic measurements of the IF in the range of non-linear magnetic DW dynamics trigger athermal variation in the static net magnetization of a sample attained after a step-like change in the applied polarizing field. The change in the net magnetization under oscillatory mechanical stress stems, in turn, from rearrangements of the DW structure which approaches an equilibrium configuration. When acoustic oscillations are excited continuously in a sample subjected to a periodic polarizing field, the ultrasonic “equilibration” of the magnetic domain structure provokes a reduction in the coercive field. 

Close to stoichiometry Ni_2_MnGa archetypal magnetic shape memory single crystals [17,18,19] were chosen as the object for experiments. This choice of material, apart from tackling the generic effect of “equilibration” of metastable magnetic DW structure by acoustic oscillations, is aimed at emphasizing the importance of sometimes neglected [20,21] magnetic domain structure in the elastic and anelastic properties of the ferromagnetic cubic phase in Ni-Mn-Ga alloys. The authors of Refs. [22,23] have revealed a significant role of magnetic domain structure in magnetoelastic effects of the cubic nearly stoichiometric Ni_2_MnGa, but no attempt has been made to analyze the magnetoelastic effects in terms of canonical components of the magnetomechanical IF impeding consistent interpretation of the field, frequency, and strain amplitude dependences of the anelastic effects in cubic ferromagnetic Ni_2_MnGa. This gap is essentially filled in Refs. [24,25]. In the present work we show that classical treatment of the magnetomechanical IF perfectly accounts for rather peculiar magnetoelastic effects in cubic Ni_2_MnGa.

## 2. Materials and Methods

Samples of nearly stoichiometric Ni_2_MnGa were prepared from the same single crystal as used in Refs. [24,25]. The crystal was produced at Adaptamat Ltd., Helsinki, Finland, annealed at 1300 K for 96 h and slowly cooled at 50 K/h. The samples oriented along [100] axis were spark cut and chemically polished. The temperatures of phase transformations were determined using ac impedance measurements: the Curie temperature *T_C_* = 383 K, the pretransformation temperature *T_PM_* = 261 K, the martensitic transformation start temperature *M_s_* = 201 K [25]. Data on magnetization and magnetic domain structure were reported in Supplement 2 of Ref. [24]. Two samples were tested and showed essentially similar results. The data for one of the samples with dimensions 11.2 × 1.1 × 1.0 mm^3^ are shown hereafter.

The IF was measured by Piezoelectric Ultrasonic Composite Oscillator Technique (PUCOT) [26], using longitudinal resonant oscillations of the samples at a frequency close to 90 kHz. A detailed description of a fully automated setup is given in Refs. [27,28]. For the samples employed, the PUCOT permits controlling the strain amplitude $\varepsilon_{0}$ of the oscillations between ca. 10^−7^ and 10^−4^. The oscillator consisting of quartz transducers and the attached sample was fixed inside the cryostat operating between 80 and 400 K. The polarizing axial magnetic field was applied to the entire oscillator by means of a long (600 mm) solenoid placed around the cryostat. The magnetic field *H* up to 12 kA/m was used. The non-uniformity of the applied field in the volume containing the sample of less than 0.5% was checked by scanning the corresponding space using Gaussmeter 475DSP (Lake Shore Cryotronics Inc., Westerville, OH, USA) equipped with a longitudinal field sensor. The homogeneity of the true magnetic field inside the sample is deteriorated by demagnetizing effects. The values of the true field are difficult to calculate, however. A reasonable approximation to estimate the demagnetizing factor is to consider thin and long sample as an ellipsoid of revolution with axial ratios of 10:1.1:1.0. Then, for the field applied along the sample, the demagnetizing factor takes a rather low value of ca. 0.02 [29]. 

The reversible inverse magnetostriction (reversible Villari effect, RVE) was studied by means of Mechanomagnetic Spectroscopy [30,31]. This modification of the PUCOT allows one to measure the RVE of magnetic materials simultaneously with the IF by picking up periodic variations of the magnetic flux density induced in a sample by periodic stress. The experimental implementation consists in placing a small pickup coil around the middle section of the sample (strain node and stress antinode for the fundamental standing wave mode). To measure the periodic RVE signal, a lock-in amplifier was used that yields the predominant first harmonic of stress-induced periodic flux density variation *B*(*t*): *B*(*t*) = *B_m_* sin (2π*ft* + *φ*) = *B*_0_ sin 2π*ft* + *B*_0_′ cos2π*ft*,(1)
where *f* is the frequency of oscillations, *B_m_* and *φ*—modulus and the phase of the RVE signal with respect to the reference strain signal, *B*_0_ and *B*_0_′—amplitudes of the real and imaginary parts, respectively, of the first harmonic of the RVE. Usually, the real part of the RVE is strongly predominant over the imaginary one, *B*_0_ ⪢ *B*_0_′. The amplitude of the first RVE harmonic, *B_m_*, and its real part *B*_0_ are used hereafter. 

Two types of experimental protocols were employed. In the first experiment, the sample was first thermally demagnetized by heating above the Curie temperature *T_C_* (up to 388 K). Then the sample was cooled down to the pre-set temperature without an applied magnetic field. After reaching and stabilization of the desired temperature, the strain amplitude dependence (SAD) of the IF was registered for the demagnetized state of the sample. After that, the polarizing magnetic field *H* was applied to the sample in a step-like manner with the step of 1.5 kA/m. For each value of *H* two consecutive strain amplitude scans were performed. 

The magnetic hysteresis of RVE was registered in the second type of experiment. The sample was again first thermally demagnetized by heating beyond *T_C_*. The RVE and IF were measured during heating and consequent cooling to the pre-set temperature under constant value of oscillatory strain amplitude without a polarizing field. Upon reaching and stabilizing the desired temperature, the RVE was measured as a function of the applied periodic polarizing field. During the measurements the oscillatory strain amplitude was kept constant at either a low value of 2 × 10^−7^ or at a high value of 10^−5^. Two cycles of the saw-tooth magnetic field were applied in each experiment. The frequency of the cyclic magnetic field was 0.001 Hz (period of 1000 s). A total of 100 experimental points were taken in each cycle of the applied field, that is, each hysteresis loop contained 100 experimental points. The RVE hysteresis was studied in the cubic phase at *T* = 343 K.

## 3. Results

Figure 1 shows SAD of the IF, expressed as logarithmic decrement δ, in Ni_2_MnGa sample taken at 343 K. The elevated temperature (well above the room temperature, 40 K below the *T_C_*) was chosen in order to reduce the saturating field and thus to arrive rather close to saturation under available moderate fields. The sample was initially heated to 388 K (above the *T_C_* ≈ 383 K) and then cooled down to 343 K under *H* = 0. A polarizing field up to *H* = 12 kA/m was then applied to the sample in eight steps of 1.5 kA/m and then reduced in the reverse sequence. Two consecutive SADs were registered in the initial state of the sample and for each value of the applied field. Each measurement of the SAD started from the initial point at a low strain amplitude $\varepsilon_{0}$. Then the forward run of the SAD was measured for increasing $\varepsilon_{0}$ up to a maximum value of ca. 2 × 10^−5^. The reverse run of the SAD was measured immediately after the forward one for the strain amplitude decreasing in the reverse sequence. Figure 1a depicts the entire set of the SADs registered for increasing *H*. The range of predominantly linear (strain amplitude-independent) IF is observed for low strain amplitudes, below $\varepsilon_{0}≅$ (2–3) × 10^−7^. Non-linearity (dependence of the IF on strain amplitude) becomes notable for $\varepsilon_{0}$ > 10^−6^ under low and moderate *H*. The subdivision into the ranges of linear and non-linear IF is conditional, however, and no “critical” strain amplitude can be found. The data in Figure 1a indicate that in the first measurements after each *H* step the forward and reverse runs of the SADs do not coincide. This effect persists for moderate *H* values up to ca. 4.5 kA/m. The difference between forward and reverse runs of the SAD is conventionally referred to as strain amplitude hysteresis [32] and is usually ascribed to a variation in the microstructure of the sample induced by mechanical oscillations. The immediate second SAD measurement for each *H* step does not show any detectable strain amplitude hysteresis. The data for the second SAD measurement coincide with the reverse run of the first SAD, pointing to the stability of the sample structure after the first SAD measurement. Figure 1a shows that, for the first measured SAD, the IF is always higher for decreasing than for increasing strain amplitude (the effect referred to as “positive hysteresis”). This difference between the IF values for increasing and decreasing strain amplitude persists both in the linear IF range and in the range of the non-linear IF behavior.

Figure 1b shows similar one-time hysteresis only in the first SAD measurement after each step-like decrease in the magnetic field. As in Figure 1a, the hysteresis is revealed for low and moderate *H*. However, details of the hysteresis are different as compared with rising *H*: in the low-amplitude linear range the IF is lower for the reverse run than for the forward one (“negative hysteresis”). Thus, the low-amplitude linear IF drops as a result of the ultrasonic excitation of material. Nevertheless, the hysteresis remains positive in the non-linear IF range. Therefore, for decreasing *H* a crossover is found from the positive hysteresis in the non-linear range to the negative hysteresis in the linear IF. Figure 1c shows that after crossing the value *H* = 0 and application of an increasing field in the opposite direction, the initial pattern of the strain amplitude hysteresis shown in Figure 1a is recovered: the hysteresis becomes positive both in linear and non-linear ranges. Thus, the SAD hysteresis induced by field variations is an even effect and does not depend on the direction of the applied magnetic field. The type of the hysteresis in the range of linear low-amplitude IF is different for increasing and decreasing *H*: the first measurement of the SAD increases the IF for rising *H* and decreases for lowering *H*.

We now analyze strain amplitude-independent low-amplitude linear $\delta_{i}$ and non-linear $\delta_{h}$ IF terms assuming, as usually, their additivity [33]. In particular, separation of magnetic losses stems from different spatial scales involved [15]. The value of the total decrement $\delta \left(\varepsilon_{0},H\right)$ is then a sum of linear $\delta_{i}$ and non-linear $\delta_{h}$ terms: (2)$$\delta \left(\varepsilon_{0},H\right)=\delta_{i}\left(H\right)+\delta_{h}\left(\varepsilon_{0},H\right).$$

Using Equation (2), the non-linear IF term $\delta_{h}\left(\varepsilon_{0},H\right).$ is derived from the experimental data as a difference between the total IF and low-amplitude IF background. Figure 2 shows $\delta_{i}$ and $\delta_{h}$ versus *H* obtained from the data of Figure 1a. The field dependence of $\delta_{i}$ was directly determined from the IF background at a low strain amplitude $\varepsilon_{0}$ = 2 × 10^−7^, marked with a vertical line in Figure 1a. The non-linear IF term was obtained as a difference between the total IF for the high strain amplitude $\varepsilon_{0}$ = 2 × 10^−5^ and the low-amplitude background (blue and black curves, respectively, in Figure 2).

A crucial observation is the existence of an intense maximum of the linear IF at *H* = 6 kA/m, concomitant with nearly complete suppression of the non-linear IF. For *H* > 6 kA/m, the linear IF diminishes rapidly and eventually drops below the initial (for *H* = 0) $\delta_{i}$ value.

Figure 3 shows a set of SADs of the non-linear IF term at 343 K for different polarizing fields. Each curve was derived from the total IF and linear IF using Equation (2). The data on log-log scale show essentially a monotonous decrease with the field, consistent with Figure 2. Over a wide range of moderate strain amplitudes, the SAD is a power law:(3)$$\delta_{h}\left(\varepsilon_{0}\right)\propto \varepsilon_{0}^{n},$$
with the exponent *n* ≅ 1, followed by a saturation.

## 4. Discussion

### 4.1. Microeddy, Macroeddy and Hysteretic Internal Friction Terms

Above the Curie temperature, both linear and non-linear IF terms fall below the background level of the quartz transducers for the same high-quality Ni_2_MnGa crystals [24]. Therefore, the IF of non-magnetic origin can be neglected in the ferromagnetic state of the cubic phase. Then the patterns of the $\varepsilon_{0}$ and *H* dependences for the linear and non-linear IF components in Figure 1, Figure 2 and Figure 3 should be analyzed in terms of three canonical components of the magnetomechanical IF [13,14,16,34,35,36,37]. These components are:
a linear microeddy current IF term, $\delta_{\mu }$, measured at low strain amplitudes and associated with individual DW displacements of much less than domain size, a linear macroeddy current IF component, $\delta_{M}$, that operates at a scale of sample size, andnon-linear hysteretic IF, $\delta_{h}\left(\varepsilon_{0}\right)$, emerging at higher strain amplitudes and related with rather large-scale displacements of DWs, comparable with domain size.

The main properties of the three magnetomechanical IF terms are as follows. 

(i)All three terms vanish at saturation.(ii)In the demagnetized state, $\delta_{\mu }$ and $\delta_{h}$ take maximum values, whereas $\delta_{M}$ is absent.(iii)The macroeddy current IF component $\delta_{M}$ passes over a maximum *versus H* at around net magnetization of (0.6–0.7) of the saturation value. (iv)The hysteretic IF versus strain amplitude is a power law $\delta_{h}\left(\varepsilon_{0}\right)\propto \varepsilon_{0}^{n},$ with the exponent *n* = 1 for moderate strain amplitudes. This regularity reflects the Rayleigh law of magnetization. At higher strain amplitudes $\delta_{h}\left(\varepsilon_{0}\right)$ passes over a maximum.

The data in Figure 1, Figure 2 and Figure 3 are in perfect agreement with points (i)–(iv). Figure 1 and Figure 2 show the expected suppression of linear and non-linear magnetomechanical IF terms by the polarizing field between 6 and 12 kA/m. These values correspond to the onset of the saturation for directly measured magnetization curves, Supplement 2 of Ref. [24]. The non-linear term declines monotonously with *H* and vanishes close to the saturation, Figure 2. 

The linear IF $\delta_{i}$ is a sum of micro- and macroeddy current components, $\delta_{\mu }$ and $\delta_{M}$, respectively:(4)$$\delta_{i}\left(H\right)=\delta_{\mu }\left(H\right)+\delta_{M}\left(H\right).$$
$\delta_{\mu }$ and $\delta_{M}$ have different field dependences: the macroeddy IF vanishes in the demagnetized state when the microeddy IF term is at maximum. Then, in the initial demagnetized state (*H* = 0) the linear IF is the microeddy current IF term as is indicated in Figure 2: (5)$${\left.\delta_{i}\right|}_{H=0}={\left.\delta_{\mu }\right|}_{H=0}.$$

Therefore, the initial curve taken at *H* = 0 kA/m is characterized by relatively low low-amplitude background and the strongest non-linearity of the SAD, Figure 1a. The applied magnetic field suppresses $\delta_{\mu }$ and promotes $\delta_{M}$. Then, observation of a monotonous rise and of a maximum of $\delta_{i}$ versus *H* points to the predominant contribution of the macroeddy term in the data for *H* ≠ 0. This ratio between micro- and macroeddy current IF terms stems from rather high frequency of microeddy relaxation in cubic Ni_2_MnGa [24]. Thus, in contrast to the non-linear hysteretic term, the decline in the microeddy current IF with *H* cannot be traced directly from the experimental data. 

Finally, the experimentally observed $\delta_{h}\left(\varepsilon_{0}\right)$ dependence (Figure 3) agrees with the Equation (3) and demonstrates the Rayleigh-type functional form of the hysteretic magnetomechanical IF with saturation in the so-called super Rayleigh range [34], representing the onset of the expected $\delta_{h}\left(\varepsilon_{0}\right)$ maximum.

### 4.2. One-Time Strain Amplitude Hysteresis of Internal Friction

The analysis performed enables us to tackle the problem of observed one-time strain amplitude hysteresis of the IF. The data obtained point to the superposition of two components of the strain amplitude hysteresis: (i) the positive hysteresis over the range of non-linear IF both for increasing and decreasing magnetic fields and (ii) the hysteresis in the linear IF range, which is positive for the increasing field and negative for the decreasing one.

Positive strain amplitude hysteresis of the magnetomechanical IF was earlier repeatedly observed, but poorly explored [14,34,38,39,40]. Adams studied this effect most carefully for a wide range of ferromagnets [40]. The one-time hysteresis (“semi-permanent” in Adams’ terminology) was observed at low and moderate oscillatory stress amplitudes and applied magnetic fields. The effect was attributed to irreversible movement of DWs through internal stress peaks, affecting the magnetization of the sample [40]. Coronel and Beshers [34] observed the strain amplitude hysteresis of the magnetomechanical IF in Armco iron, which was accompanied by partial recovery of the IF with time on decreasing strain amplitude. They explained this hysteresis by a dispersal of carbon atoms trapped by DWs and their subsequent return by diffusion. Coronel and Beshers noted also that some of the increased IF can be removed only by demagnetization. This non-recoverable with time part of the hysteresis was ascribed to changing magnetic state after high-amplitude oscillations [34].

To begin with we mention that since the observed hysteresis is a one-time effect, it is not associated with any time dependence of the IF, contrary to the classic observations initiated by Chambers and Smoluchowski [32]. Therefore, the hysteresis is not related with any diffusion-controlled redistribution of obstacles by oscillating DWs. 

Equation (4) indicates that the origin of the hysteresis in the linear IF is in the variation in microeddy and/or macroeddy current IF terms, provoked by a combined action of an applied magnetic field and ultrasonic oscillations in the non-linear IF range. The first important point is that the non-linear IF is related with the relatively large-scale motion of non-180° magnetic DWs, trapped in the metastable states after the step-like increase/decrease in the polarizing field. Therefore, the oscillatory motion of the DWs in the range of non-linear anelasticity provokes their transition into deeper energy minima closer to the equilibrium state. This “equilibration” of the metastable DW structure by high-amplitude ultrasonic oscillations is a feasible interpretation of the strain amplitude hysteresis. The key issue of the change in the type of hysteresis from positive to negative for increasing and decreasing fields is then easily explained. Indeed, after increasing/decreasing the field, DWs are trapped by the obstacles that impede DW motion towards the equilibrium state. Obviously, the equilibrium magnetization value is higher than the one attained for the increasing field and lower for the decreasing field. Triggering of the DW irreversible motion to the equilibrium state by acoustic oscillations thus provokes the increase in the net magnetization of the sample for the series of increasing *H* steps and the decrease in the net magnetization for the series of decreasing *H* steps. The change in the net magnetization (for low and moderate fields) provokes, in its turn, the change in the macroeddy current IF. This is the reason for the ultrasonically induced increase in $\delta_{i}$ for increasing *H* steps and decrease for decreasing *H* steps and of the strain amplitude hysteresis of different signs. The second and the following measurements of the SAD do not produce any additional magnetization change, accounting for the one-time property of the strain amplitude hysteresis.

As for the positive hysteresis over the range of non-linear IF, it is clearly related with the hysteretic component of the magnetomechanical IF. This hysteresis indicates that, independently of the sign of magnetic field change, DWs become more mobile in their new positions after the irreversible motion to the equilibrium state. This means that acoustic oscillations in the non-linear IF range shake DWs off the obstacles trapping them during unidirectional motion after the step-like increase/decrease in the polarizing field.

The suggested interpretation of strain amplitude hysteresis is supported by its observation only at moderate values of increasing and decreasing *H*. Studies of the Barkhausen noise intensity (see e.g., [41]) indicate that this range corresponds to the most intense rearrangement of DW structure. Strain amplitude hysteresis vanishes with an increasing field in parallel with the suppression of the non-linear hysteretic IF when the DW density diminishes. This is the *H* range wherein the net magnetization of the sample becomes progressively controlled by the rotation of magnetization and the hysteresis in magnetization versus *H* curves vanishes.

### 4.3. Effect of Ultrasonic Oscillations on Magnetoelastic Hysteresis

The suggested interpretation of the observed strain amplitude hysteresis implies additional activation of the DW mobility by their stress-induced oscillations in the non-linear range. Therefore, one should expect narrower magnetic hysteresis when non-linear oscillations of DWs are excited in the sample as compared to the hysteresis registered without oscillations. To check this prediction we performed mechanomagnetic spectroscopy tests with the same sample. This method yields the hysteresis of reversible inverse magnetostriction (reversible Villari effect, RVE) versus *H*.

Since zero RVE points correspond to zero net magnetization of the sample [12], the width of the RVE hysteresis yields a distance between the *H* points wherein the net magnetization of the sample vanishes, i.e., the coercive force *H_c_*. Thus, to verify our interpretation, we compare the widths of the hysteresis loops RVE versus *H* with and without non-linear oscillations.

To confirm the magnetic origin of the signal captured by the pickup coil, we performed measurements of the amplitude of the first harmonic of the pickup coil signal versus temperature crossing the Curie temperature *T_C_*. If the signal has the magnetic origin it should vanish in the paramagnetic state of the material. Figure 4 shows the temperature spectra of the amplitude of the first harmonic of the *B_m_* given by Equation (1) in the vicinity of *T_C_*. The oscillatory strain amplitude was maintained at 2 × 10^−7^ and a magnetic field was not applied, *H* = 0. The sample was initially magnetized by the field of 12 kA/m applied at 343 K.

Data in Figure 4 point to an abrupt drop of the magnetic signal on heating at around the Curie temperature *T_C_* ≈ 383 K. The signal becomes negligible above *T_C_* proving that the coil signal represents the RVE. Upon cooling, the RVE signal rises at *T_C_*, although the polarizing field was not applied and RVE should vanish for zero net magnetization. It can be shown, however, that the low RVE signal registered on cooling under *H* = 0 is due to the weak uncompensated magnetic field of the Earth. 

Figure 5 shows the hysteresis loops “real part of the first harmonic of RVE, *B*_0_, versus *H*″ registered at 343 K. The hysteretic loops were recorded for low oscillatory strain amplitude (belonging to the range of small DW displacements in the linear mode, $\varepsilon_{0}$ = 2 × 10^−7^) and for high strain amplitude ($\varepsilon_{0}$ = 10^−5^, well in the range of non-linear DW dynamics). The hysteresis loops registered with a maximum polarizing field of *H* = 12 kA/m are presented in Figure 5a. Figure 5b displays the results of the experiments for which the maximum applied field was limited to 4.5 kA/m. This experiment permitted resolving details of the RVE behavior close to *H* = 0. Figure 5c represents the central parts of the two hysteresis loops from Figure 5b on an expanded scale. For clarity, only the hysteresis loops registered in the second cycle of the applied magnetic field are shown in Figure 5. 

Several observations deserve mentioning as far as the general features of the hysteresis loops are concerned. First, the RVE shows a maximum at ca. 5 kA/m and declines at higher applied fields towards zero values. The decline in the RVE for *H* > 5 kA/m is a sign of magnetic saturation [12]. The position of the RVE maximum is therefore close to the maximum of $\delta_{M}$, Figure 2. Second, it is quite remarkable that the hysteresis loops are jerky over the intermediate range of the applied field and become smooth both at a low applied field and close to saturation. Third, each hysteresis loop shows three sections separated by cross-over points of changing the direction of circulation along the loop. Each branch of the hysteresis loops has two cross-over points at about 1 and 6.5 kA/m. Thus, the direction of the circulation along the loop changes twice up to the saturating field. The low-field and high-field sections are characterized by smooth shape and counter-clockwise circulation. The shape of the intermediate sections of the hysteresis loops is less regular (jerky) and shows clockwise direction of circulation. The unusual direction of circulation along the hysteresis loop is allowed in mixed variables, since the area of the loop does not represent dissipated energy. The variation in the circulation for mixed variables is reported for Ni-Fe-Ga ferromagnetic shape memory alloy [42] and various ferroelectrics [43]. The data available are not sufficient to interpret these features of the hysteresis loops and our goal is merely to compare the width of the loops with and without high-amplitude ultrasonic excitation. 

Although the loops are rather narrow, Figure 5c indicates that the hysteresis loop registered with high-amplitude excitation is located inside the hysteresis loop under low-amplitude excitation. This experiment confirms our prediction of a more equilibrium state of DWs when high-amplitude ultrasonic oscillations are excited in a sample subjected to a periodic magnetic field. Thus, the oscillatory stress-induced relaxation of magnetic DWs closer to equilibrium is a one-time effect during SAD measurements after a step-like increase in the polarizing field and becomes a continuous phenomenon during RVE tests under a magnetic field ramp. The relaxation of DW configuration can be reformulated in terms of the net magnetization of the sample: the superposition of ultrasonic oscillations additionally increase the net magnetization under a raising field and reduce it when the applied field diminishes.

### 4.4. Stress Anisotropy Field

The effect of acoustic oscillations on mechanical deformation of crystals is well known. Conventional dislocation plasticity is enhanced by superimposed oscillations. This phenomenon is represented by the so-called Blaha-Langeneker or acoustoplastic effect [44,45]. During active deformation, acoustic oscillations reduce the flow stress due to the extra plastic strain induced by superposition of acoustic oscillations on the quasistatic stress. Acoustic oscillations also promote pseudoelastic strain during direct and reverse stress-induced martensitic transformation (acousto-pseudoelastic effect) [46]. A consequence of the acousto-pseudoelastic effect is a reduction in the stress–pseudoelastic strain hysteresis in Cu-Al-Ni single crystals [46]. Both acoustoplastic and acousto-pseudoelastic effects involve an additional strain under a combined action of static and oscillatory stress. In the present work we report a fundamentally similar effect, but observed for mixed variables: the superposition of oscillatory mechanical stress and static magnetic field affects the magnetization of the sample. 

Mixed variables involved in the uncovered effect can be reduced to conjugate ones using basics of the magnetoelastic coupling phenomenon. As was mentioned in the Introduction, the magnetomechanical IF is also a manifestation of such coupling related to the stress-induced motion of magnetic DWs and, hence, dissipation of energy. The equivalence between the applied stress and magnetic field is expressed using a so-called stress anisotropy field. For an isotropic ferromagnet with saturation magnetization $M_{s}$ and magnetostriction $\lambda_{s}$, the applied stress σ creates a stress anisotropy field $H_{\sigma }$ [14]:(6)$$H_{\sigma }≅\frac{3}{2}\frac{\sigma \lambda_{s}}{\mu_{0}M_{s}},$$
where $\mu_{0}$ is the permeability of the vacuum.

The isotropic approximation is quite legitimate for cubic Ni_2_MnGa single crystals, which have very low magnetocrystalline anisotropy and DW structure characteristic of amorphous materials (Supplement 2 of Ref. [24]). The stress anisotropy field superimposed on the static field *H* thus creates the magnetization change Δ*M*:(7)$$\Delta M=\Delta M\left(H,H_{\sigma }\right).$$

Equations (6) and (7) yield the phenomenological interpretation of the uncovered effect. An important issue raised in the microscopic interpretation of the acoustoplastic and acousto-pseudoelastic effects is related to the possibility to apply the superposition principle to the ultrasonic and quasistatic stresses [46,47]. Experimental data available point to the existence of the critical stresses in the acoustoplastic and acousto-pseudoelastic effects that might prove that the superposition principle is not applicable at low oscillatory stress amplitudes. The reason of the failure of superposition is ascribed to the necessity for the ultrasonic oscillations to be intense enough (in other words, to have sufficient amplitude) to assist the dislocations (conventional plasticity) or twin boundaries (pseudoelastic strain in transformation plasticity) in overcoming obstacles impeding their mobility to contribute to the additional macroscopic strain. Therefore, the occurrence of the acoustoplastic and acousto-pseudoelastic effects is restricted to the range of strain amplitudes wherein anelasticity is strongly non-linear. 

The present results are not sufficient to contribute to the similar analysis of the uncovered effect. Nevertheless, it is instructive to estimate the values of the stress anisotropy field for the cubic Ni_2_MnGa. We rewrite Equation (6) for the anisotropy field amplitude $H_{\sigma_{0}}$ using experimental values of the oscillatory strain amplitude $\varepsilon_{0}$: (8)$$H_{\sigma_{0}}≅\frac{3}{2}\frac{\varepsilon_{0}E\lambda_{s}}{\mu_{0}M_{s}},$$
where *E* is the Young′s modulus. 

We estimate $H_{\sigma_{0}}$ for two values of strain amplitude: $\varepsilon_{0}$ = 2 × 10^−7^ and $\varepsilon_{0}$ = 10^−5^. For *E* ≈ 10 GPa [24], $\lambda_{s}$ ≈ 50 × 10^−6^ [48] (note that the magnetostriction $\lambda_{s}$ is negative), $\mu_{0}M_{s}$ ≈ 0.5 T (Supplement 2 of Ref. [24]), a negligible value $H_{\sigma_{0}}$ ≈ 0.3 A/m is obtained for $\varepsilon_{0}$ = 2 × 10^−7^. The strain amplitude $\varepsilon_{0}$ = 10^−5^ yields $H_{\sigma_{0}}$ ≈ 15 A/m. This value is two orders of magnitude lower than the applied field step of 1.5 kA/m for the experiments in Figure 1. Therefore, assuming the linear dependence of macroeddy (more precisely, of the sum of micro- and macroeddy) IF on the applied field, Figure 2, one should expect the magnitude of the strain amplitude hysteresis to be approximately 0.01 of the IF variation induced by the field step. The data in Figure 1 indicate that the magnitude of the hysteresis is an order of magnitude higher. This analysis supports the hypothesis that the ultrasonic oscillations activate the irreversible motion of DWs trapped in local energy minima. Therefore, the effect of ultrasonic oscillations is stronger than simple superposition of the static field *H* and periodic stress anisotropy field $H_{\sigma_{0}}$.

## 5. Conclusions

One-time strain amplitude hysteresis of the magnetomechanical internal friction provoked by step-like variations of the polarizing field is found in single crystalline samples of Ni_2_MnGa ferromagnetic shape memory alloy in the cubic phase. It is shown that the hysteresis consists of two components. The hysteresis in the linear internal friction range, whose sign is different for increasing and decreasing fields, is due to the macroeddy component of magnetomechanical internal friction, controlled by the net magnetization of the sample. This hysteresis is associated with additional increase/decrease in magnetization that stems from relaxation of magnetic domain walls trapped in metastable states. The relaxation is triggered by ultrasonic oscillations. The hysteresis in the non-linear internal friction range is due to the hysteretic component of magnetomechanical internal friction. It is positive independently of the sign of magnetic field change, indicating increased mobility of magnetic domain walls after their acoustically-induced relaxation. 

The suggested interpretation is verified using measurements of the magnetic field hysteresis of the reversible inverse magnetostriction (reversible Villari effect). It is shown that application of high-amplitude ultrasonic oscillations reduces the width of magnetic hysteresis. 

The present work reports the generic effect for ferromagnetics. On the other hand, experiments with functional Ni_2_MnGa magnetic shape memory material confirm that peculiarities of its elastic and anelastic properties in the cubic ferromagnetic state are dominated by often neglected magnetic domains and domain walls.

## Figures and Tables

**Figure 1 materials-16-02547-f001:**
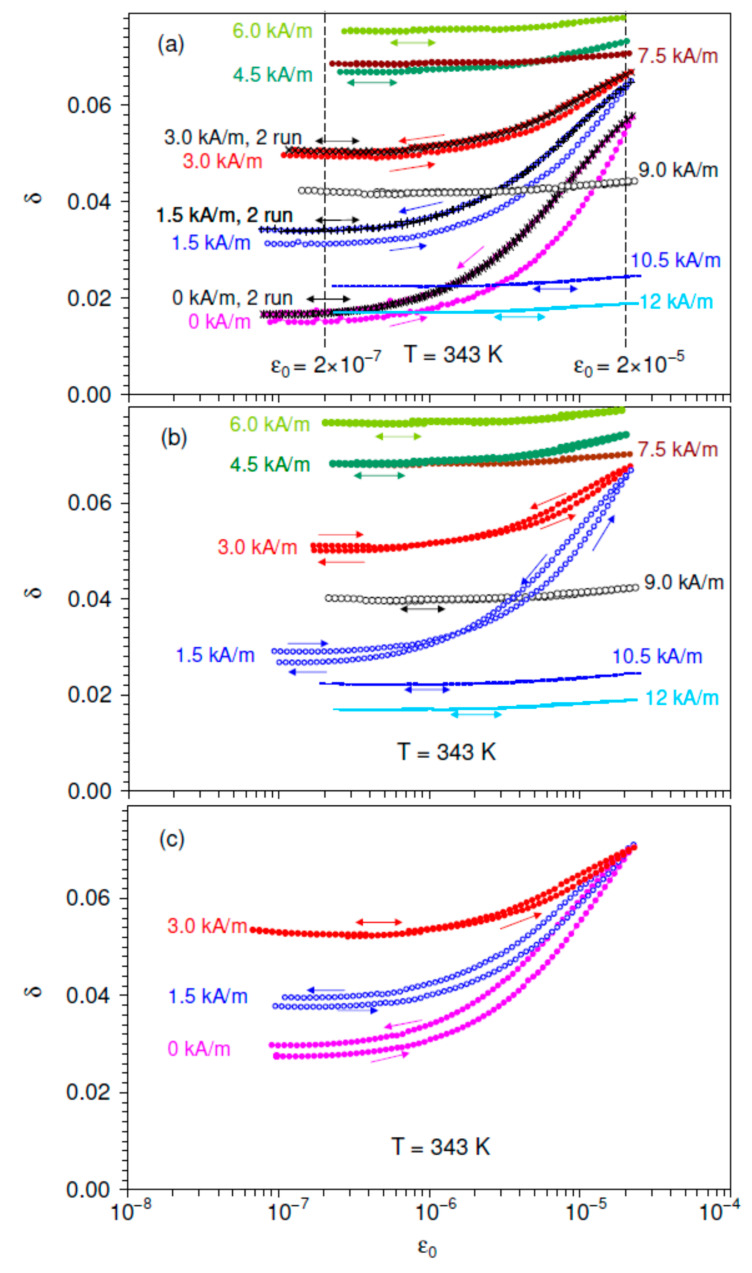
Strain amplitude dependence of the internal friction δ for single crystalline Ni_2_MnGa sample at 343 K under different values of increasing (**a**), decreasing (**b**), and increasing in the opposite direction (**c**) polarizing magnetic field. The first measurement is shown for each value of the applied field, except for the field of 0, 1.5, 3.0 kA/m for increasing field (**a**), where the data for two consecutive runs are shown. The arrows indicate the direction of the strain amplitude variation. Dotted vertical lines in (**a**) mark the low and high strain amplitudes $\varepsilon_{0}$ = 2 × 10^−7^ and $\varepsilon_{0}$ = 2 × 10^−5^ for which the field dependence of the linear, total, and non-linear internal friction terms were determined.

**Figure 2 materials-16-02547-f002:**
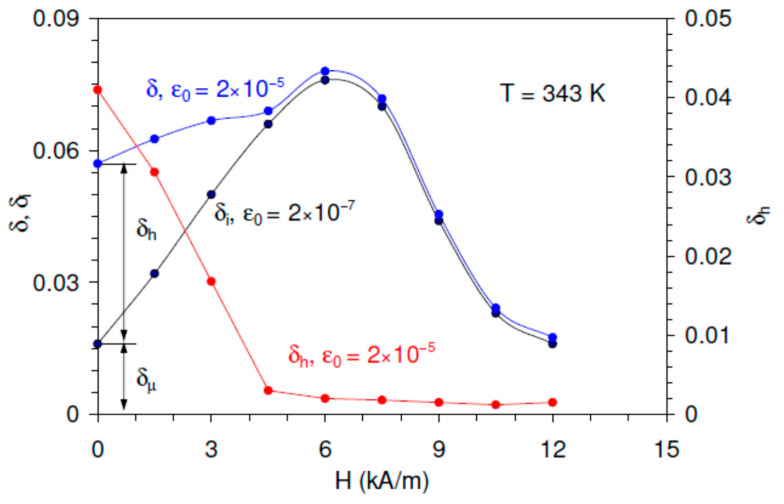
The total δ and linear $\delta_{i}$ internal friction at high and low strain amplitudes and their difference, the non-linear component $\delta_{h}$ of the total internal friction, versus polarizing field *H* for temperature *T* = 343 K. The linear internal friction is determined for the low strain amplitude $\varepsilon_{0}$ = 2 × 10^−7^, the total and non-linear internal friction for the high strain amplitude $\varepsilon_{0}$ = 2 × 10^−5^. The decomposition of the total internal friction for the demagnetized state at *H* = 0 into the linear microeddy current $\delta_{\mu }$ and non-linear hysteretic $\delta_{h}$ internal friction is shown.

**Figure 3 materials-16-02547-f003:**
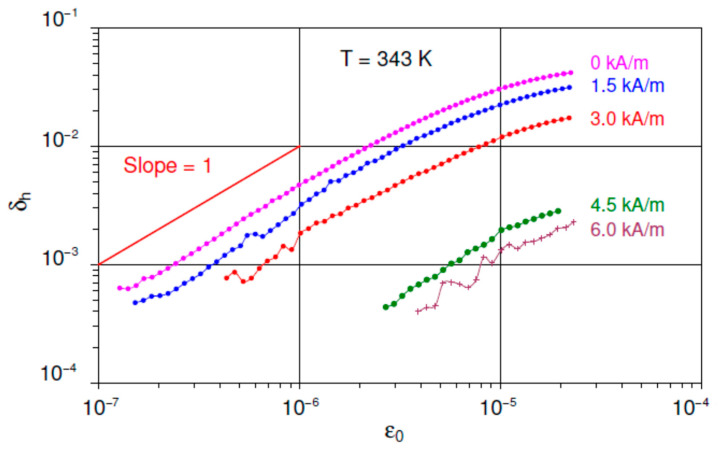
Strain amplitude dependent part of the internal friction $\delta_{h}$ for Ni_2_MnGa sample at 343 K versus strain amplitude $\varepsilon_{0}$ under different values of increasing the polarizing magnetic field. The data are shown for increasing strain amplitudes during the first run for each value of the applied field.

**Figure 4 materials-16-02547-f004:**
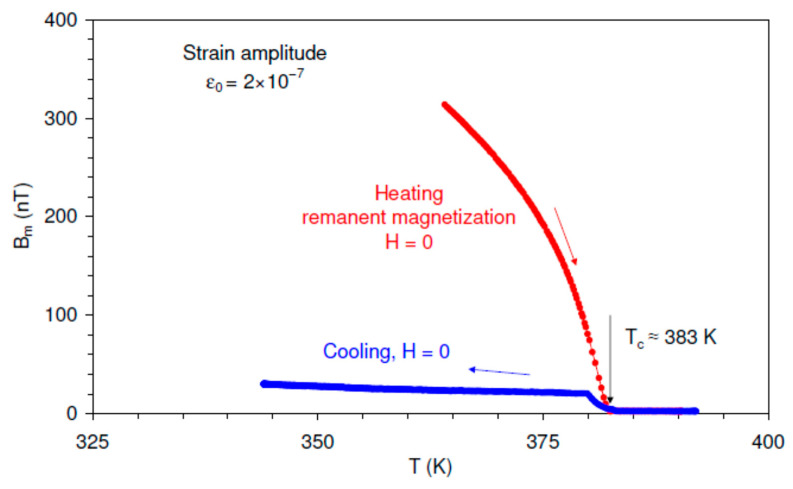
Temperature spectra, close to the Curie temperature *T_C_* ≈ 383 K, of the amplitude of the reversible inverse magnetostriction (reversible Villari effect) *B_m_* for the Ni_2_MnGa sample on heating in the remanent state and on cooling from above the *T_C_* under zero applied field. Oscillatory strain amplitude $\varepsilon_{0}$ = 2 × 10^−7^.

**Figure 5 materials-16-02547-f005:**
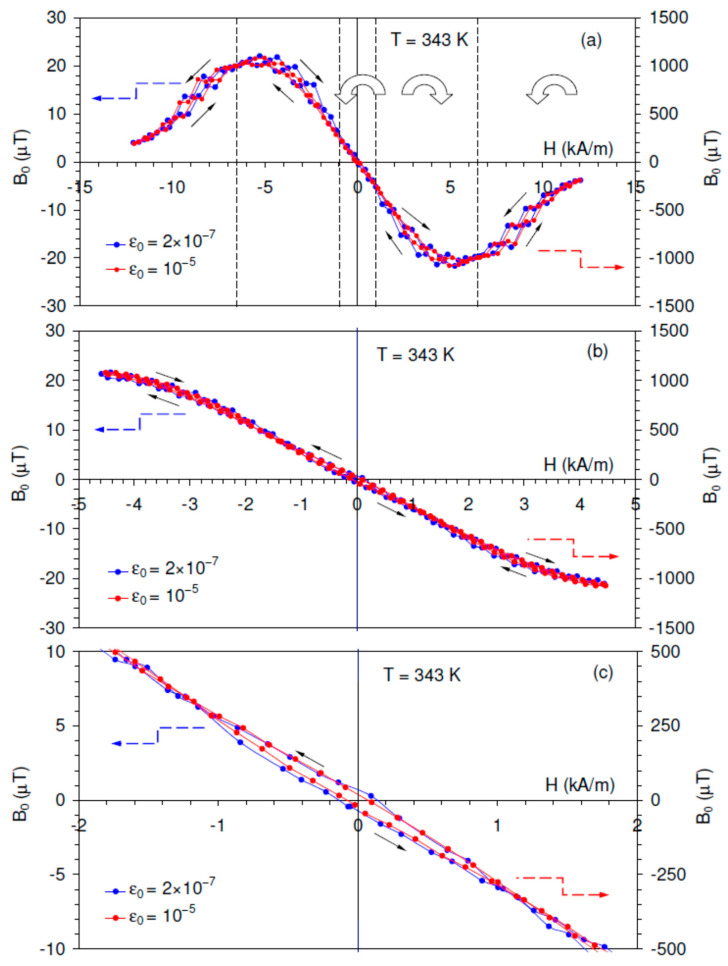
Real part of the first harmonic of reversible inverse magnetostriction *B*_0_ (reversible Villari effect) of the Ni_2_MnGa sample versus cyclic polarizing field (frequency of the cycle −0.001 Hz). Measurement at 343 K at two different strain amplitudes $\varepsilon_{0}$ = 2 × 10^−7^ and $\varepsilon_{0}$ = 10^−5^: (**a**) hysteresis loops for the field *H* between −12 and 12 kA/m; dotted vertical lines mark the cross-over points wherein the direction of the circulation along the loop changes. Arrows mark the direction of circulation along the hysteresis loops; (**b**) hysteresis loops for the field *H* variation between −4.5 and 4.5 kA/m; (**c**) central parts of the loops from (**b**) on an expanded scale. The hysteresis loops correspond to the second cycle of the applied magnetic field.

## Data Availability

The data presented in this study are available in the article.

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
