# Peer review of "Effect of Acoustic Oscillations on Non-Equilibrium State of Magnetic Domain Structure in Cubic Ni2MnGa Single Crystal"

_materials, 2023, doi:10.3390/ma16072547_

Round 1
Reviewer 1 Report
Dear Ms. Serena Shi, the team of the Materials Editorial Office,
Thank you for the review invitation from the Materials journal.
The Manuscript ID: materials-2289105, titled " Effect of Acoustic Oscillations on Non-equilibrium State of Magnetic Domain Structure in Cubic Ni2MnGa Single Crystal".
The manuscript is interesting, well-planned, and well-written. However, the current work requires a minor revision before it can be published in the Materials journal.
The authors have been provided with a list of suggestions for enhancing the article in the Comments and Suggestions for Authors section, which I hope they will consider.
The author/s discussed the Impact of acoustic oscillations on the Non-equilibrium state of the magnetic domain structure in a single cubic Ni2MnGa crystal. Moreover, the authors focused on the importance of magnetic domain structure in the elastic and anelastic properties of the ferromagnetic cubic phase in Ni-Mn-Ga alloys. However, presenting and discussing the results seems compelling, and the current work requires a minor revision before it can be published in the Materials journal.
Some major suggestions have been made to improve the article as follows:
1- The manuscript should be carefully checked and corrected for grammatical, typographical, and punctuation errors.
2- The abstract section looks like part of an introduction, so it is better to rephrase it to highlight the most essential obtained results.
3- The introduction lacks the problem statement to which the author will contribute to finding appropriate solutions. As a suggestion: The problem should be stated, and the solution method used in the current study should be described. Moreover, the author should refer to recent literature reviews to indicate the priority of the current work.
4- The analysis of the study's findings and the data presentation seem to have been carefully planned and well-reported. To demonstrate the importance of the present study, it could be required to compare the investigated results with those of the previous ones.
5- Many cited works appear outdated, so an updated list of references is advised.
Best wishes,
Jamal M. Rzaij
Author Response
Response to Reviewer 1 Comments
Point 1. The manuscript should be carefully checked and corrected for grammatical, typographical, and punctuation errors.
Response 1. The manuscript has been extensively edited in order to avoid misuses and misprints and to make the text more readable.
Point 2. The abstract section looks like part of an introduction, so it is better to rephrase it to highlight the most essential obtained results.
Response 2. The abstract has been modified by including additional information on the experimental results and their interpretation. The authors believe that the abstract represents the essence of all sections of the manuscript: Introduction (as has been correctly commented by the Reviewer), Experimental methods and materials, Results, Discussion and Conclusions. It might be that the relative “weight” of the Introduction section is somewhat higher than that of the other sections of the manuscript, due to the first phrase of the Abstract. Still we believe that this potential disbalance is really minor and eliminating the first phrase would make the brief description of the content of the manuscript in the Abstract much less clear.
Point 3. The introduction lacks the problem statement to which the author will contribute to finding appropriate solutions. As a suggestion: The problem should be stated, and the solution method used in the current study should be described. Moreover, the author should refer to recent literature reviews to indicate the priority of the current work.
Response 3. We are grateful to the Reviewer for this comment. We would like to mention that, on one hand, classical concepts of magnetomechanical internal friction are known for decades. Therefore, the references are given in the manuscript to the best known reviews, textbooks and original works on magnetomechanical effects by Bozorth, Bertotti, Beshers, Nowick and Berry, Degauque, etc. On the other hand, we believe that the experimental results and the analysis of magnetomechanical effects in cubic Ni2MnGa presented in Refs. 24, 25 and in the present work are pioneering for this functional material. Therefore, any review article dealing with application of the classical concepts to Ni2MnGa might not exist; at least the present authors are not aware of such a publication.
A paragraph has been added in the Introduction section, referring to recent publications which deal with the magnetoelastic effects in the cubic ferromagnetic Ni2MnGa but remain short in application of the analysis based on the classical pattern of magnetomechanical effects in conductive ferromagnets. We would appreciate if the Reviewer could kindly provide us with relevant recent publications that might have been overlooked.
Point 4. The analysis of the study's findings and the data presentation seem to have been carefully planned and well-reported. To demonstrate the importance of the present study, it could be required to compare the investigated results with those of the previous ones.
Point 5. Many cited works appear outdated, so an updated list of references is advised.
Response 4 and 5.
A paragraph in the final part of the Introduction has been added which deals with a couple of recent publications wherein the effect of magnetic domain structure in the properties of the cubic Ni2MnGa is mentioned.
The authors are surprised that such an important structural element of a ferromagnetic material as magnetic domains and domain walls is still generally disregarded, even in rather advanced publications, see e.g. Refs. 20,21. The lack of the analysis of the elastic and anelastic properties of ferromagnetic cubic Ni2MnGa is related to the necessity of having experimental facilities that would permit reliable separating of linear and non-linear anelastic terms, since these terms possess quite different (sometimes opposite) frequency and magnetic field dependences. In addition, both linear terms of the magnetomechanical internal friction become negligible at low frequencies. The ultrasonic experimental technique employed in the present work seems to be a rare example of optimal method for investigations of magnetic domain-related effects.
We would be grateful to the Reviewer for any reference to further update the list of references.
Reviewer 2 Report
In this paper, the authors investigate the effect of acoustic oscillations on the magnetic domain structure of a cubic Ni2MnGa single crystal in a non-equilibrium state. After step-like variations in the polarizing magnetic field, the magnetomechanical absorption of ultrasound vs strain amplitude was analyzed. During step-like fluctuations in the polarizing field, hysteresis was discovered in the strain amplitude dependences of magnetomechanical internal friction. In addition, the observed equilibration of the magnetic domain structure caused by acoustic oscillations is linked to the periodic stress anisotropy field induced by oscillatory mechanical stress. The work sounds excellent and displays some ingenuity.
The new version of this work needs to address a few key areas before submission.
1. I feel that the introduction might use a bit more time invested on it. We greatly appreciate it when authors update their works cited to include the most recent publication.
2. There has to be a strong focus in the updated text on the unique contributions of the current work.
3. Whilst the abstract itself is fine, the authors should expand on the data and commentary presented there.
4. The authors would do well to provide more information in the experimental section (Materials and Methods) about the materials they really used (Cubic Ni2MnGa Single Crystal)
5. The revised manuscript could benefit from the inclusion of material characterization analyses specific to the used material (Cubic Ni2MnGa Single Crystal). These may include, XRD, FTIR, SEM and TEM.
6. There are just a few typos and grammatical errors discovered throughout the text; however, the writers are respectfully requested to verify the entire corrected text for any remaining errors.
7. The graphical abstract is also appreciated for the current work. Since it will provide the reader with a good impression of the substance of the text, it is highly appreciated that the authors designed a creative and engaging graphical abstract for this work. To put it another way, the writers may include it within the discussion section of the paper as an illustrative scheme.
8. Finally, it is recommended that the conclusion section be written in paragraph form rather than using bullet points. The most important findings must also be included in the section that is devoted to drawing conclusions.
Author Response
Point 1. I feel that the introduction might use a bit more time invested on it. We greatly appreciate it when authors update their works cited to include the most recent publication.
Point 2. There has to be a strong focus in the updated text on the unique contributions of the current work.
Response 1 and 2.
We have added a paragraph to the Introduction section in which we provide a couple of references that arrive to the conclusion on the necessity of dealing with the magnetic domains in the cubic ferromagnetic phase of Ni2MnGa. However, these works fail to advance due to the lack of experimental facilities and, hence, the data that would permit analysis of different mechanisms of the magnetomechanical anelastic effects. Probably even basic understanding of the standard pattern of magnetomechanical internal friction is missing in these works. We add the reference [24] to the introduction, in which an example of the appropriate analysis was given. The authors believe that it is not necessary to repeat such analysis in the present work and focus the attention on the fact that the canonical treatment of the magnetomechanical internal friction explains rather curious and unusual phenomena reported.
Point 3. Whilst the abstract itself is fine, the authors should expand on the data and commentary presented there.
Response 3. The abstract has been modified. The modified version includes more references to the experimental results and their interpretation.
Point 4. The authors would do well to provide more information in the experimental section (Materials and Methods) about the materials they really used (cubic Ni2MnGa single crystal).
Point 5. The revised manuscript could benefit from the inclusion of material characterization analyses specific to the used material (cubic Ni2MnGa single crystal). These may include, XRD, FTIR, SEM and TEM.
Response 4 and 5.
Ni-Mn-Ga system is by far the most studied magnetic shape memory alloy, with thousands of publications and well documented properties. While Ref. 17 deals with the first report of the magnetic field induced strain in Ni-Mn-Ga, Refs. 18 and 19 are extensive reviews on the properties of these alloys. The specific single crystals used in the present work were previously exhaustively studied by XRD, SEM, AFM, MFM, etc., and reported in previous publications (some of them with the present authors). The major part of the corresponding properties is not directly related to the present results, however, and is not included in the manuscript. We use the Ref. 24, Supplement 2, where the data on magnetic domain structure and magnetization curves were reported for the same single crystals. These data are the most relevant for the present work, but has already been published and cannot be reproduced again.
Point 6. There are just a few typos and grammatical errors discovered throughout the text; however, the writers are respectfully requested to verify the entire corrected text for any remaining errors.
Response 6. The text of the manuscript has been extensively revised and, hopefully, becomes more readable now.
Point 7. The graphical abstract is also appreciated for the current work. Since it will provide the reader with a good impression of the substance of the text, it is highly appreciated that the authors designed a creative and engaging graphical abstract for this work. To put it another way, the writers may include it within the discussion section of the paper as an illustrative scheme.
Response 7. We did not understand well this proposal. In other journals, the graphical abstract is, indeed, required for reviewing purposes. This is not the case of “Materials”, however. It is not clear also how it can be included in the Discussion section. We would prefer to keep the general style and organization of this section as is.
Point 8. Finally, it is recommended that the conclusion section be written in paragraph form rather than using bullet points. The most important findings must also be included in the section that is devoted to drawing conclusions.
Response 8. We thank the Reviewer for this suggestion. This restructuring has been done. We would like to mention that sometimes the referees do not accept presentation of the experimental findings in the section of Conclusions. This seems to be a matter of style/taste and we would prefer to focus the short and compact Conclusions section essentially on the true conclusions drawn from the experimental observations.
Reviewer 3 Report
1- How did you check that the applied magnetic field is homogeneous.
2-What is the orientation of the magnetic field ( parrallel or perpendicular to the sample. Is there a difference in the results?).
3-How do you explain that the curve in figure 5 (c) does not pass through zero.
4-what kind of anisotropy field you describe in the paper ( of shape, crystalline, etc ) or the total anisotropy?
Author Response
Point 1. How did you check that the applied magnetic field is homogeneous.
Response 1. The appropriate paragraph has been added in the manuscript (page 3, lines 107-114). The homogeneity of the applied external field was checked by scanning the corresponding space using a magnetic field sensor. The homogeneity of the true field in the sample is, indeed, quite difficult to estimate. However, we argue in the revised version that the demagnetizing factor for the axial field is very small for the thin and long sample used and assume that the distortions of the true field can be disregarded.
Point 2. What is the orientation of the magnetic field (parrallel or perpendicular to the sample. Is there a difference in the results?).
Response 2. It is stated in the manuscript that the field is applied in the axial direction, along the sample. The results will be qualitatively the same if the transverse field is applied, but all dependences will be shifted to higher external field due to the demagnetizing effect.
Point 3. How do you explain that the curve in figure 5 (c) does not pass through zero.
Response 3. Thanks for this comment. These curves correspond to the second cycle of applied field. This detail has been added both in the text and in the figure caption.
Point 4. What kind of anisotropy field you describe in the paper ( of shape, crystalline, etc ) or the total anisotropy?
Response 4. It is indicated in the text on several occasions that we deal with the magnetic anisotropy induced by applied stress, which can be described in terms of the stress anisotropy field.